# Effects of Season of Birth and Meteorological Parameters on Serum Bilirubin Levels during the Early Neonatal Period: A Retrospective Chart Review

**DOI:** 10.3390/ijerph18052763

**Published:** 2021-03-09

**Authors:** Shigeo Iijima, Toru Baba, Miyuki Kondo, Tomoka Fujita, Akira Ohishi

**Affiliations:** Department of Pediatrics, Hamamatsu University School of Medicine, Hamamatsu 4313192, Japan; tbb0727@yahoo.co.jp (T.B.); mulberry714@gmail.com (M.K.); rasp_berry_tm103@yahoo.co.jp (T.F.); a-ohishi@hama-med.ac.jp (A.O.)

**Keywords:** neonatal jaundice, bilirubin, season, meteorological parameter, temperature, sex

## Abstract

To establish whether serum bilirubin levels vary in healthy term neonates according to seasonal variations and meteorological factors, we retrospectively studied 3344 healthy term neonates born between 2013 and 2018. Total serum bilirubin (TSB) levels were measured on the fourth day after birth. The monthly and seasonal variations in TSB levels and clinical and meteorological effects on TSB levels were assessed. In the enrolled neonates, the median TSB level was 195 µmol/L. The TSB level peaked in December and was the lowest in July, but the variation was not statistically significant. The TSB level was significantly higher in the cold (October to March) than in the warm season (April to September; *p* = 0.01). The comparison between seasonal differences according to sex showed TSB levels were significantly higher in the cold than in the warm season in male infants (*p* = 0.001), whereas no significant difference was observed in female infants. A weakly negative but significant association existed between TSB levels and the mean daily air temperature (r = −0.07, *p* = 0.007) in only the male population; the female population showed no significant correlation between TSB levels and meteorological parameters. The season of birth is an etiological factor in neonatal jaundice, with an additional influence from sex.

## 1. Introduction

Neonatal jaundice is characterized by yellow discoloration of the skin and sclera of the newborn due to the accumulation of unconjugated bilirubin. Most neonates develop physiological jaundice because of increased bilirubin levels in the blood, which is due to the combined effect of high red cell turnover, immature hepatic conjugation, and enhanced resorption of bilirubin by the enterohepatic circulation [1]. Neonatal hyperbilirubinemia is associated with a variety of conditions and manifests in approximately 60% of full-term neonates and almost all preterm neonates, with a prevalence greater than 80% [1]. Although hyperbilirubinemia is usually a benign physiologic condition, very high bilirubin levels occur in certain pathologic conditions and may cause an injury to the central nervous system, called kernicterus spectrum syndrome. The severity of jaundice varies between infants and may be associated with factors such as race, sex, nutrition, eating habits, hormones, and genetic factors [1,2,3,4].

In addition, clinical experience suggests that seasonal variations influence the occurrence and severity of neonatal hyperbilirubinemia. The relationship between the season of birth and physical parameters may depend on environmental effects during the critical period of an infant’s development [5]. An important event occurring soon after birth is the steady increase in serum bilirubin levels, with recent studies suggesting that bilirubin has a protective effect against secondary oxidants that the neonate is exposed to at birth [2,6]. Several previous studies have found that the development of neonatal jaundice and hyperbilirubinemia depends on the season of the neonate’s birth [7,8,9,10,11,12,13,14,15,16,17,18]. However, there is no consensus on the influence of the season of birth on the risk of developing neonatal jaundice.

The purpose of this study was to identify differences in neonatal jaundice according to sex and season of birth. We conducted a retrospective review of data collected from a large population over a long time period and analyzed the total serum bilirubin (TSB) levels in four-day-old healthy term neonates to determine the presence of monthly and seasonal variations in TSB levels as well as the association with meteorological parameters.

## 2. Materials and Methods

### 2.1. Subjects and Data Collection

We conducted a retrospective chart review of consecutive healthy term neonates who were born in Hamamatsu University Hospital between 1 January 2013 and 31 December 2018. Only Japanese neonates were included in our study to minimize the effects of confounding variables. To evaluate the TSB levels in healthy neonates, we excluded neonates with low birthweight (<2500 g), macrosomia (≥4000 g), those delivered preterm (<37 weeks) or post-term (≥42 weeks), and those admitted to the neonatal intensive care unit. In addition, we excluded neonates with underlying conditions likely to cause hyperbilirubinemia, such as blood type incompatibility, glucose-6-phosphate dehydrogenase deficiency, intrauterine infection (toxoplasma, rubella, cytomegalovirus, herpes, syphilis, and human immunodeficiency virus), major congenital malformations, clinical syndromes, and chromosomal abnormalities. Based on the findings from previous studies [3,12,19,20], data on sex, birthweight, gestational age, delivery mode, perinatal asphyxia, nutrition, weight loss in the first 4 days (from birth to pre-discharge check-up), and hematocrit level after birth were obtained as potential predictors of TSB levels.

### 2.2. Ethics Approval

The present study was designed and conducted in accordance with the general principles outlined in the Declaration of Helsinki and was approved by the ethics committee of Hamamatsu University School of Medicine (approval number: 19–103). For this retrospective study, the parents of the subjects were not required to provide informed consent because the analysis used anonymous clinical data that had been obtained after each patient’s parent provided written consent for clinical management. Moreover, we provided an opt-out method for study consent on our hospital website.

### 2.3. Blood Sampling and Total Serum Bilirubin and Hematocrit Measurements

Capillary whole blood samples for the measurement of TSB and hematocrit levels were routinely obtained from a heel prick during the day 4 check-up. After a physical examination, a single heel prick was performed, and a whole blood sample (approximately 40 µL) was collected in a heparinized capillary tube for the measurement of TSB and hematocrit levels. The blood samples were transported to the laboratory located a few meters away from the newborn nursery at room temperature, and the TSB level was measured within few minutes of blood sampling by the optical density method using a Bilmeter F (Mochida-Siemens, Tokyo, Japan); the measurement error of the device was ±3% according to the manufacturer’s report. Hematocrit levels were measured using the manual microhematocrit method, a gold standard method for hematocrit determination. Blood sampling and bilirubin and hematocrit measurements were conducted by six experienced neonatologists who were trained in using the Bilmeter F.

In our hospital, every neonate received care in the newborn nursery, which was air-conditioned at approximately 26 °C throughout the day. All evaluations began at 10:00 a.m. and were performed in the newborn nursery.

### 2.4. Collection of Seasonal Data and Meteorological Parameters

Japan is an island country running from north to south along East Asia’s Pacific coast, and Hamamatsu City is in the central region (latitude 34.7° north). The climate is relatively mild, with an annual average temperature of 16.6 °C. For this study, the seasons were divided into a warm season (April to September) and a cold season (October to March). The meteorological data on the day the neonates were born were obtained from the Japan Meteorological Agency website [21], and the mean air temperature, mean relative humidity, mean barometric pressure, total precipitation amount and sunshine duration were noted. These data were collected at the Hamamatsu Local Meteorological Observatory, which is located approximately 1 mile from the hospital.

### 2.5. Statistical Analysis

Data are presented as medians with interquartile ranges or as means ± standard deviations for continuous variables and as counts and percentages for categorical variables. Non-parametric methods (Spearman correlation coefficient, Kruskal–Wallis test, and the Mann–Whitney U test) were used to assess the influence of clinical, seasonal, and meteorological parameters on TSB levels. A linear regression model was built using TSB levels and meteorological parameters with significant seasonal variation as dependent and independent variables, respectively. Data were analyzed using the Statistical Package for Social Sciences (SPSS version 25, IBM, Tokyo, Japan). All statistical tests were two-sided, and a *p*-value < 0.05 was considered statistically significant.

## 3. Results

### 3.1. Samples

During the study period, a total of 4772 neonates were born in our hospital. After excluding those meeting the exclusion criteria and those with missing or unreliable data, 3344 neonates were included in our study for analysis (Figure 1). In the enrolled neonates (male/female ratio, 1685:1659), the median birthweight was 3025 g, and the median gestational age was 39.4 weeks. The median TSB level was 195 µmol/L (interquartile range, 168–227 µmol/L). Overall, 1698 neonates were delivered during the warm season, and 1646 were delivered during the cold season. There were no significant differences in the birthweight, gestational age, delivery mode, 1 min and 5 min Apgar scores, nutrition, weight loss in the first four days after birth, or hematocrit levels between infants born in the warm and cold seasons (Table 1). Sex was associated with significant differences in birthweight (*p* = 0.03), gestational age (*p* = 0.04), nutrition (*p* < 0.001), and weight loss in the first four days after birth (*p* < 0.001; Table 1). TSB levels were slightly higher in male infants than in female infants, although the difference was not statistically significant.

### 3.2. Monthly and Seasonal Variations in the Meteorological Parameters

The mean daily air temperature was the lowest in January (median, 6.0 °C) and peaked in August (median, 28.0 °C). The mean daily relative humidity peaked in July (median, 76.0%) and was the lowest in February (median, 51.5%). The mean daily barometric pressure was the lowest in August (median, 1002.6 hPa) and peaked in December (median, 1012.6 hPa). The total daily precipitation amount peaked in September (mean, 11.9 mm) and was the lowest in January (mean, 1.7 mm). The daily duration of sunshine was lowest in September (median, 4.2 h) and peaked in May (median, 8.9 h). A comparison of the climatic variables between the warm and cold seasons indicated that the air temperature, relative humidity, and levels of precipitation were significantly higher in the warm than in the cold season (all *p* < 0.001). In contrast, the barometric pressure was significantly lower in the warm than in the cold season (*p* < 0.001). The duration of sunshine was slightly higher in the warm than in the cold season, but this difference was not statistically significant.

### 3.3. Monthly and Seasonal Variations in Total Serum Bilirubin Levels

On a monthly basis, the TSB level peaked in December (median, 202 µmol/L) and was the lowest in July (median, 190 µmol/L; Figure 2), although the variation was not statistically significant. In male infants, the TSB level was the highest in December (median, 209 µmol/L) and the lowest in July (median, 190 µmol/L), which was found to be a significant difference (*p* = 0.04). Conversely, in female infants, the TSB level was the highest in March (median, 200 µmol/L) and the lowest in May (median, 188 µmol/L), but without significant variation. The TSB levels were significantly higher in the cold than in the warm season (*p* = 0.01; Table 1). When we compared seasonal differences according to sex, the TSB levels were significantly higher in the cold season in male infants (*p* = 0.001), whereas the difference was not significant in female infants (Table 2).

### 3.4. Correlation between Total Serum Bilirubin Levels and Meteorological Parameters at Birth

Simple linear regression analysis showed that there were no significant correlations between TSB levels and the mean daily air temperature (*p* = 0.23), mean daily relative humidity (*p* = 0.17), mean daily barometric pressure (*p* = 0.08), daily total precipitation amount (*p* = 0.06), and daily duration of sunshine (*p* = 0.06). Only in the male population, there were weakly negative but significant correlations between TSB levels and the mean daily air temperature (r = −0.07, *p* = 0.007); in addition, the TSB levels had a weakly positive but significant correlation with the mean daily barometric pressure (r = 0.06, *p* = 0.01) and daily total precipitation amount (r = 0.07, *p* = 0.008). Stepwise multiple linear regression analysis was conducted to evaluate the impact of the independent variables (mean daily air temperature, mean daily barometric pressure, and total daily precipitation amount) on TSB levels. Only the mean daily air temperature was found to significantly influence the TSB levels (*p* = 0.04). When analyzing only the female population, simple linear regression analysis showed no significant correlation between TSB levels and any of the climate parameters.

## 4. Discussion

In the present study, TSB levels were significantly higher in infants born during the cold season (October to March) than those born during the warm season. Only in the male population, the TSB level was found to have a weak but significant negative correlation with the mean daily air temperature at birth. Therefore, the season of birth appears to be an etiological factor in neonatal jaundice. Furthermore, TSB levels also appear to be influenced by sex.

In 1969, Milby et al. described seasonal variations in the incidence of neonatal hyperbilirubinemia for the first time [7]. In their study, the incidence of neonatal unconjugated hyperbilirubinemia was significantly higher during the fourth quarter of each year. However, climatic information was not described, and the cause of the seasonal fluctuation was unclear. Subsequently, twelve other studies have been conducted to investigate the possible impact of the season of birth on the serum bilirubin levels of neonates or pathological hyperbilirubinemia [8,9,10,11,12,13,14,15,16,17,18,22] (Table 3). Two of these studies demonstrated that TSB levels were higher in the cold than in the warm season, in agreement with the present study [9,22], although the study by Hojat et al. did not confirm the presence of a statistically significant difference. Anttolainen et al. suggested that the short duration of daylight experienced during the cold season could increase the incidence of hyperbilirubinemia [8]. Sunlight can prevent hyperbilirubinemia [23] because the sun emits blue-green light in the spectrum needed to most effectively convert bilirubin to its water-soluble isomers for excretion. Hojat et al. also described that there is less sunlight in the winter, and the decomposition of bilirubin decreases during this time [22]. In addition, they suggested that parents often increase a baby’s room temperature to prevent hypothermia in winter, and this can cause dehydration and increase serum bilirubin levels. In the present study, there was no statistical significance in the seasonal and monthly variations of sunshine duration, and no significant correlation between serum bilirubin level and sunshine duration was found. In contrast, nine previous studies have demonstrated that serum bilirubin levels and the incidence of pathologic hyperbilirubinemia were significantly higher in the warm than in the cold season [8,11,12,13,14,15,16,17,18] (Table 3); González et al. suggested that high temperatures during the summer and the associated higher dehydration rate may be the main cause [11]. Additionally, they suggested that the seasonal differences could be due to breastmilk jaundice. Ahmady et al. also suggested that, during the summer season, the increased temperature led to increased breastfeeding rates to compensate for dehydration and elevated bilirubin levels [17]. Breastfeeding has been recognized as a contributing factor for the development of neonatal hyperbilirubinemia, as the breastmilk of some women contains a metabolite of progesterone called 3α,20β-pregnanediol, which inhibits UDP-glucuronosyltransferase (UGT) bilirubin glucuronidation activity [4]. Moreover, breastfed babies, particularly those who have difficulty nursing or getting adequate nutrition from breastfeeding, are at a higher risk of jaundice. Besides dehydration, a low caloric intake through inadequate levels of breastfeeding may contribute to the onset of jaundice. In the present study, the room temperature was not included in the evaluated environmental factors, and the daily feeding volume was also not investigated. However, the weight loss rate in the first four days after birth was not significantly different between the warm and cold seasons. Therefore, dehydration of the newborn during the warm season is unlikely. Regarding the room temperature, our previously published paper reported that the room temperature in the neonatal ward was unchanged throughout the year [24]. Moreover, in the present study, the rate of breastfeeding of neonates was not different between those born in the warm and cold seasons. Scrafford et al. found that the high ambient temperatures increased the risk of neonatal jaundice because the family’s reluctance to be outdoors due to extreme heat reduced the potential for phototherapy [16].

In the present study, we observed that meteorological conditions affected the TSB levels in neonates who were maintained at a controlled temperature all day. Moreover, the subjects in the present study were never exposed to outdoor temperatures following birth. Animals are programmed by the ambient temperature during the prenatal period to adapt their organs to postnatal changes in ambient temperature, probably due to epigenetic changes in gene regulation [25]. We speculate that, in humans, seasonal and meteorological variations during pregnancy may have a significant effect on the adaptability of neonates to their environment following birth.

Regarding sex differences, eight of thirteen studies refer to sex having a role in the seasonal variation of bilirubin levels [7,8,10,12,14,16,17,18] (Table 3). Seven of them reported that serum bilirubin levels or the incidence of pathological hyperbilirubinemia were higher in male infants than in female infants [7,8,10,14,16,17,18]. Moreover, Bottini et al. reported that the rise of serum bilirubin levels in the warm season was significantly higher in female infants than in male infants, whereas in the cold season, no significant differences between male and female infants were observed [12]. In addition, two studies reported that conjugated bilirubin was more common in female infants than in male infants [13,14]. Sex divergent glucuronidation rates were observed in humans, and sex differences in UGT mRNA have been observed in animal studies [26,27]. Sex hormones may be an important regulator of conjugation. In a previous study, the phenomenon of protection from oxidative stress was shown to be more marked in male than female newborns [10].

Several other clinical factors may influence neonatal hyperbilirubinemia. Considering other factors may generate new criteria for the management of physiological and pathological neonatal jaundice. As bilirubin has a protective effect against secondary oxidative stress [2,6], seasonal variation of birth stress may influence bilirubin levels during the first few days of life [12]. In one study, the highest number of births leading to cerebral palsy occurred in spring, with the lowest number occurring in winter [28]. Low birthweight and premature birth are well recognized as major risk factors for exaggerated hyperbilirubinemia [1]. Seasonal patterns of low birthweight and preterm births have been found [29]. As pregnant women are particularly sensitive to meteorological conditions and environmental exposure [5], the period before delivery could be a critical window influencing fetal growth when high or low ambient temperature exposure occurs [29]. In summer, heat stress can damage antioxidant defense systems and lead to increased oxytocin secretion [30]. In winter, decreased sunlight exposure may lead to lower levels of vitamin D [31], which is essential for normal placental function and fetal growth [32]. The mode of delivery may also influence the jaundice risk. In previous studies, lower bilirubin levels were observed after cesarean section (CS), and this was supposedly explained by placental transfusion or the timing of cord clamping [33,34]. However, other studies comparing CS with vaginal delivery did not show a difference in hyperbilirubinemia risk [19,35]. As most bilirubin is derived from hemoglobin degradation [20], serum bilirubin level may increase as hemoglobin or hematocrit level increases. In the present study, the birthweight, gestational age, 1 min and 5 min Apgar scores (as indicators of birth stress), delivery mode, and hematocrit level were not associated with significant seasonal variations. Apart from clinical factors, genetic factors have also been associated with neonatal jaundice [4]. Considerable daily and seasonal variations in serum bilirubin levels have been observed in patients with Gilbert’s syndrome (GS), a benign deficiency in bilirubin glucuronidation [36]. GS is associated with a homozygous polymorphism, A(TA)7TAA, in the TATA-box of the promoter region of the bilirubin UGT1A1 gene [37]. Neonates with such a variant have accelerated jaundice progression during the first two days of life. In Caucasians, the (TA)7 allele is important because its allele frequency is 30–40% [38]. In Japanese, the (TA)7 polymorphism is less frequent, although the Gly71Arg mutation may contribute to neonatal hyperbilirubinemia [39]. In the present study, genetic markers were not investigated because genetic testing was not routinely conducted in neonates with physiological or pathological jaundice.

The present study had limitations. First, the retrospective study design restricted the appropriate assessment of potential confounders to determine cause and effect. Second, the present study analyzed only TSB levels because direct and indirect bilirubin levels were not measured during routine bilirubin evaluation in normal neonates. Both Ahmady et al. and Bala et al. demonstrated that direct bilirubin levels were elevated in the winter season [17,18]. Analysis of direct and indirect bilirubin levels may provide new insights into understanding the seasonal effects of TSB in our study. Third, the meteorological data used for statistical analyses were not available at the time when the neonates were born or when their blood was sampled. In addition, data on the indoor temperature in the maternity ward were not available. For mothers, personal exposure to meteorological indicators may have been modified by the duration of time spent indoors before delivery. The actual exposure of individuals to meteorological conditions might not always be the same as the recorded data of a specific geographical region. As a result, these may lead to some degree of evaluation error.

The present study also has strengths. Information about seasonal variation in neonatal jaundice is limited because most other studies are based on small sample sizes, a limited number of variables, and short study periods. Unlike many previous studies, this study evaluated the association between serum bilirubin levels and meteorological data in the early neonatal period with a relatively large sample size.

## 5. Conclusions

The present study evaluated the relationship between season of birth and neonatal jaundice and the hypothesis that such a relationship is mediated at least in part by meteorological factors and sex. Our findings suggest that the season of birth is an etiological factor in neonatal jaundice, with an additional influence of meteorological factors and sex. However, the contradictions observed in various studies and the mechanisms of seasonal fluctuations in serum bilirubin levels remain unclear. There might be other and hitherto unknown mechanisms masquerading as seasonal variations. Further large, prospective studies are required to address this important issue.

## Figures and Tables

**Figure 1 ijerph-18-02763-f001:**
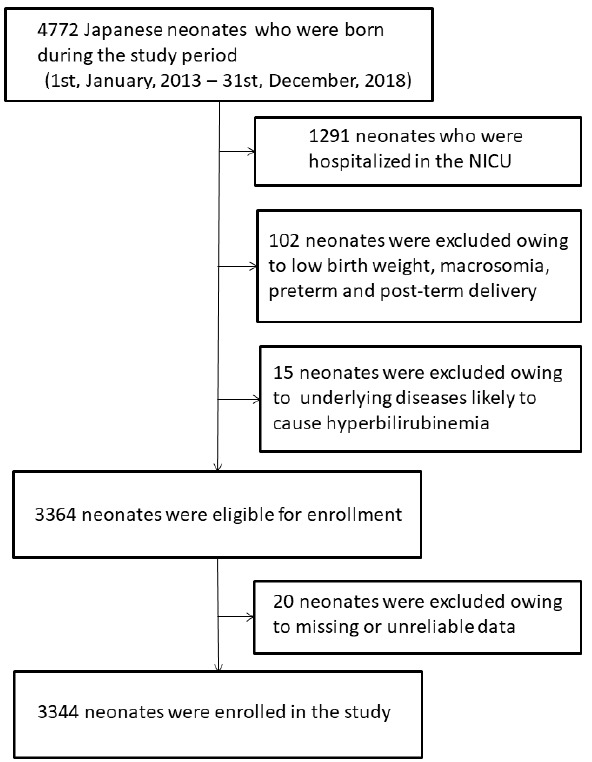
Flow chart of subject recruitment. NICU: neonatal intensive care unit.

**Figure 2 ijerph-18-02763-f002:**
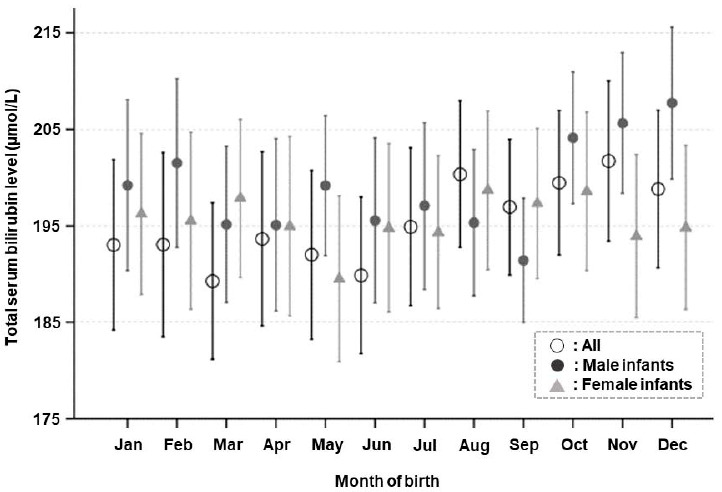
Serum bilirubin levels by the month of birth. Mean and 95% confidence interval for monthly total serum bilirubin level in healthy term neonates at four days after birth.

**Table 1 ijerph-18-02763-t001:** Seasonal and sex differences in the characteristics of the study population and meteorological parameters.

Parameter	Season			Sex		
	Warm Season *n* = 1698	Cold Season *n* = 1646	*p*-Value	Male *n* = 1685	Female *n* = 1659	*p*-Value
Clinical parameters						
Birthweight, g	3022 (2810–3256)	3030 (2824–3280)	0.27	3034 (2836–3280)	3018 (2800–3251)	0.03
Gestational age, weeks	39.3 (38.4–40.1)	39.4 (38.4–40.3)	0.06	39.3 (38.4–40.1)	39.4 (38.4–40.2)	0.04
Male sex, (%)	848 (49.9)	837 (50.8)	0.60	-	-	-
Warm season, (%)	-	-	-	848 (50.3)	850 (51.2)	0.60
Delivery mode (vaginal delivery:cesarean section), (%)	1348:350 (79:21)	1338:308 (81:19)	0.90	1351:334 (80:20)	1335:324 (80:20)	0.08
Apgar score at 1 min	8 (8–9)	8 (8–9)	0.87	8 (8–9)	8 (8–9)	0.57
Apgar score at 5 min	9 (9–9)	9 (9–9)	0.89	9 (9–9)	9 (9–9)	0.88
Nutrition (breast:mix:formula), (%)	181:1478:39 (11:87:2)	162:1459:25 (10:89:1)	0.93	241:1400:44 (14:83:3)	102:1537:20 (6:93:1)	<0.001
Weight loss in the first four days after birth, %	2.6 (1.0–4.1)	2.6 (1.0–4.3)	0.38	2.4 (1.0–4.1)	2.7 (1.1–4.3)	0.04
Hematocrit level, %	52.1 ± 6.0	52.2 ± 5.7	0.57	52.2 ± 5.8	52.0 ± 5.9	0.26
Total serum bilirubin level, µmol/L	193 (166–226)	197 (169–231)	0.01	195 (169–229)	193 (166–227)	0.05
Meteorological parameters						
Daily mean air temperature, °C	23.3 (20.2–26.9)	10.4 (6.9–15.5)	<0.001	18.4 (10.5–23.6)	18.4 (10.1–23.5)	0.42
Daily mean relative humidity, %	75 (68–83)	60 (51–72)	<0.001	70 (58–79)	70 (57–79)	0.77
Daily mean barometric pressure, hPa	1006 (1002–1009)	1012 (1008–1016)	<0.001	1008 (1004–1013)	1008 (1004–1013)	0.75
Daily total precipitation amount, mm	6.8 ± 17.2	4.0 ± 12.5	<0.001	5.5 ± 16.3	5.3 ± 13.8	0.74
Daily sunshine duration, h	6.6 (1.6–10.7)	7.7 (2.6–9.5)	0.06	7.0 (2.0–9.8)	7.3 (1.9–9.8)	0.67

Categorical variables are expressed as value (%); continuous variables are expressed as mean ± standard deviation or median (interquartile range).

**Table 2 ijerph-18-02763-t002:** Seasonal differences in the characteristics of the study population and meteorological parameters according to sex.

Parameter	Male Sex				Female Sex			
	Total *n* = 1685	Warm Season *n* = 848	Cold Season *n* = 837	*p*-Value	Total *n* = 1659	Warm Season *n* = 850	Cold Season *n* = 809	*p*-Value
Clinical parameters								
Birthweight, g	3034 (2836–3280)	3022 (2830–3270)	3044 (2840–3290)	0.37	3018 (2800–3251)	3022 (2792–3232)	3016 (2810–3266)	0.53
Gestational age, weeks	39.3 (38.4–40.1)	39.3 (38.4–40.0)	39.4 (38.4–40.1)	0.13	39.4 (38.4–40.2)	39.4 (38.4–40.1)	39.4 (38.4–40.3)	0.26
Delivery mode (vaginal delivery:cesarean section), (%)	1351:334 (80:20)	673:175 (79:21)	678:159 (81:19)	0.14	1335:324 (80:20)	675:175 (79:21)	660:149 (82:18)	0.08
Apgar score at 1 min	8 (8–9)	8 (8–9)	8 (8–9)	0.49	8 (8–9)	8 (8–9)	8 (8–9)	0.34
Apgar score at 5 min	9 (9–9)	9 (9–9)	9 (9–9)	0.83	9 (9–9)	9 (9–9)	9 (9–9)	0.67
Nutrition (breast:mix:formula), (%)	241:1400:44 (14:83:3)	126:694:28 (15:82:3)	115:706:16 (14:84:2)	0.96	102:1537:20 (6:93:1)	55:784:11 (7:92:1)	47:753:9 (6:93:1)	0.71
Weight loss in the first 4 days after birth, %	2.4 (1.0–4.1)	2.4 (0.9–4.1)	2.5 (1.0–4.1)	0.22	2.7 (1.1–4.3)	2.7 (1.1–4.2)	2.7 (1.1–4.4)	0.98
Hematocrit level, %	52.2 ± 5.8	52.2 ± 6.1	52.2 ± 5.6	0.95	52.0 ± 5.9	51.9 ± 5.9	52.1 ± 5.9	0.47
Total serum bilirubin level, µmol/L	195 (169–229)	191 (166–224)	200 (174–233)	0.001	193 (166–227)	193 (166–227)	195 (166–227)	0.75
Meteorological parameters								
Daily mean air temperature, °C	18.4 (10.5–23.6)	23.4 (20.5–26.9)	10.5 (6.9–15.6)	<0.001	18.4 (10.1–23.5)	23.2 (19.7–26.9)	10.0 (6.8–15.2)	<0.001
Daily mean relative humidity, %	70 (58–79)	75 (67–82)	60 (52–72)	<0.001	70 (57–79)	75 (68–83)	60 (51–71)	<0.001
Daily mean barometric pressure, hPa	1008 (1004–1012)	1005 (1002–1009)	1012 (1008–1016)	<0.001	1008 (1004–1013)	1006 (1002–1009)	1012 (1008–1015)	<0.001
Daily total precipitation amount, mm	5.5 ± 16.3	6.9 ± 19.2	4.1 ± 12.6	<0.001	5.3 ± 13.8	6.6 ± 15.0	3.8 ± 12.3	<0.001
Daily sunshine duration, h	7.0 (2.0–9.8)	6.2 (1.8–10.6)	7.6 (2.5–9.5)	0.30	7.3 (1.9–9.8)	6.8 (1.3–10.8)	7.7 (2.9–9.5)	0.10

Categorical variables are expressed as value (%); continuous variables are expressed as mean ± standard deviation or median (interquartile range).

**Table 3 ijerph-18-02763-t003:** Reports investigating the seasonal variation in serum bilirubin levels and hyperbilirubinemia in neonates.

Study and Country	Study Type	Inclusion Criteria	Sample Size	Age at Study	Seasonality	Sex Difference	Summary
Milby et al., 1969, USA [7]	R	Serum indirect bilirubin, >171 µmol/L	170	Unknown	Yes	Yes	The incidence of hyperbilirubinemia was high during the fourth quarter of each year. Elevated bilirubin levels were more frequent in male than in female infants.
Lee et al., 1970, China [8]	P	Serum total bilirubin, ≥256 µmol/L	2687	Unknown	Yes	Yes	The incidence of neonatal jaundice was significantly higher in summer (April–September) and among male infants than in winter (October–March) and among female infants, respectively.
Anttolainen et al., 1975, Finland [9]	P	BW, <2500 g	76	0–10 d	Yes	NA	Total bilirubin levels were lower from March to August than from September to February.
Friedman et al., 1978, UK [10]	P	Unknown	12,461	0–7 d	Yes	Yes	The incidence of neonatal jaundice had monthly fluctuations. The incidence of neonatal jaundice and peak plasma bilirubin levels were significantly higher in males than in females.
González et al., 1996, Spain [11]	P	Serum total bilirubin, >274 µmol/L	161	Unknown	Yes	NA	Pathologic hyperbilirubinemia was more common in fall and less common in winter (not a statistically significant difference), while that due to breastmilk jaundice was significantly more common in summer.
Bottini et al., 2000, Italy [12]	P	Unknown	343	23–25 h	Yes	Yes	The rise of serum bilirubin was minimal in autumn (September–November). The rise of serum bilirubin showed a tendency to decrease from winter to autumn. The rise of serum bilirubin levels in spring and summer was significantly higher in female infants than in male infants.
Ding et al., 2001, China [13]	P	GA, ≥37 weeks BW ≥ 2500 g	875	1–7 d	Yes	NA	The mean peak of serum bilirubin was higher in summer than in winter.
Bottini et al., 2003, Italy [14]	P	Unknown	5540	Unknown	Yes	Yes	The maximum incidence of phototherapy was observed from May to August. The proportion of infants undergoing phototherapy was lower in females than in male infants.
Cerna et al., 2010, Czech Republic [15]	R	Healthy, term	565	Unknown	Yes	NA	The incidence of hyperbilirubinemia was higher in summer. The frequency of phototherapy use was higher during summer.
Scrafford et al., 2013, Nepal [16]	P	Unknown	18,985	Unknown	Yes	Yes	An increased risk of neonatal jaundice was observed among infants born in the hot season (March–October). A significant 3% increase in the risk of neonatal jaundice was indicated for each 1 °C increase in ambient temperature on the infant’s birth date. Male sex was a high-risk factor for neonatal jaundice.
Ahmady et al., 2015, Egypt [17]	P	GA, 37 weeks	500	Unknown	Yes	Yes	Total and unconjugated bilirubin levels were higher in newborns born in summer than in those born in winter. Increased unconjugated bilirubin level was common in male infants born in summer. Total and unconjugated bilirubin levels in female infants were significantly raised in summer. Conjugated bilirubin level was elevated in winter.
Bala et al., 2015, India [18]	P	GA, 37 weeks	1000	Unknown	Yes	Yes	Total and indirect bilirubin levels were higher in summer than in winter. Indirect bilirubin level was high in male infants in summer. In female infants, the total and indirect bilirubin levels were significantly high in winter. Direct bilirubin level was high in winter.
Hojat et al., 2018, Iran [22]	P	GA, 38–42 weeks BW, 2500–3400 g	400	7 d	No	NA	The highest bilirubin level was observed in winter (not statistically significant).

R, retrospective study; P, prospective study; GA, gestational age; BW, birthweight; NA, not analyzed.

## Data Availability

The datasets supporting the conclusions of this article are included within the article.

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
