# Peer review of "Effects of Season of Birth and Meteorological Parameters on Serum Bilirubin Levels during the Early Neonatal Period: A Retrospective Chart Review"

_ijerph, 2021, doi:10.3390/ijerph18052763_

Round 1

Reviewer 1 Report

This is an interesting and well written paper on a topic that has not received much attention. The authors have found several papers that I had not been aware of, so cudos to them. On the other hand, there are a handful of relevant papers that they might wish to consider in a revision before publication:

Ding G et al in Chin Med J 2001;114:344-7.

Lee KH et al in J Obstet Gynaecol Br Commonw 1970;77:561-4. doi: 10.1111/j.1471-0528.1970.tb03568.x.

Scrafford CG et al in Trop Med Int Health 2013;18:1317-28. doi: 10.1111/tmi.12189

Friedman L et al in BMJ 1978;1:1235-7

Of these, the Scrafford paper from Nepal is the one that has the largest population n of >18000

I think the authors' discussion is appropriate in that the mechanism behind the apparent seasonal variation in neonatal jaundice has no obvious explanation, and the gender differences as well as the peaks in hot weather in some reports and in cold weather, as in the present paper, are puzzling. It makes you wonder whether there is some other and hitherto unknown mechanism "masquerading" as seasonal.

A very minor comment is that the authors use the term "kernicterus", while recent suggestions for an updated terminology use the term "kernicterus spectrum syndrome".

Author Response

Thank you for your comments and suggestions; the manuscript has been revised accordingly. Revisions made to the manuscript are indicated with red-font text, and we have provided point-by-point responses to your suggestions below.

  1. The authors have found several papers that I had not been aware of, so cudos to them. On the other hand, there are a handful of relevant papers that they might wish to consider in a revision before publication:

Ding G et al in Chin Med J 2001;114:344-7.

Lee KH et al in J Obstet Gynaecol Br Commonw 1970;77:561-4.

Scrafford CG et al in Trop Med Int Health 2013;18:1317-28

Friedman L et al in BMJ 1978;1:1235-7.

Of these, the Scrafford paper from Nepal is the one that has the largest population n of >18000

Answer:

Thank you for your suggestions and valuable references. Despite our extensive literature review, we had missed these papers. We have now added information from the above studies in Table 3 and in the Introduction and Discussion sections.

Page no. 9 (lines 199 and 214); page no. 10 (lines 236–238); page no. 11–12 (Table 3); page no. 13 (lines 250 and 252).

  1. I think the authors’ discussion is appropriate in that the mechanism behind the apparent seasonal variation in neonatal jaundice has no obvious explanation, and the gender differences as well as the peaks in hot weather in some reports and in cold weather, as in the present paper, are puzzling. It makes you wonder whether there is some other and hitherto unknown mechanism “masquerading” as seasonal.

Answer:

Thank you for your comment; we completely agree with you. We have added a sentence in accordance with your suggestion in the Conclusions section.

Page no. 14 (lines 322–323).

  1. A very minor comment is that the authors use the term “kernicterus”, while recent suggestions for an uploaded terminology use the term “kernicterus spectrum syndrome”.

Answer:

Thank you for your comment. According to your suggestion, we have changed the term “kernicterus” to “kernicterus spectrum syndrome.”

Page no. 2 (lines 43–44).

Reviewer 2 Report

 Several studies raise the possibility of an internal seasonal clock in humans that provides an endocrine regulation for reproduction, growth, metabolism, and stress adaptation in winter−spring transition. Animal studies show these changes with a circannual rhythm even when maintained in constant photoperiod and temperature conditions. It would be important to focus this transition.

In the context of seasonal  fisiological changes blood parameters were also extensively studied. Taking in acount that most of plasma bilirubin results from the catabolism of hemoglobin (Hb), it is expected that bilirubin concentration will increase as Hb concentration increases. The Hb concentration and red blood mass also explain the  diferences in TSB serum levels observed between males and females. It would be important to analyse hematological parameters of the neonates since they also apear to change across the seasons.

The study should also have taken into account the analysis of  direct and indirect bilirubin levels. It would  bring new clues in understanding this  seasonal changes.

One of the most important factors that affect total serum bilirrubin (TSB) levels in the caucasian population is the presence  of the (TA)alelle (Gilbert Syndrome). The frequency of the (TA)7 allele is 30–40% in Caucasians, while it is less frequent in the Japanese population, with the frequency being approximately 16%. It has been suggested that the GLY71Arg variant and other polymorphisms may contribute to Gilbert's syndrome in Asian patients. To date, published data confirm the presence of GS contributes to neonatal hyperbilirubinemia, resulting in an important confouder factor to be taken into account.

It is always important to consider additional factors that may affect serum levels of bilirubin , which may generate new criteria for the diagnosis and treatment of physiological and non-physiological neonatal jaundice.

Author Response

Thank you for your suggestions; the manuscript has been revised accordingly. Revisions made to the manuscript are indicated with red-font text, and we have provided point-by-point responses to your suggestions below.

  1. In the context of seasonal fisiological changes blood parameters were also extensively studied. Taking in acount that most of plasma bilirubin results from the catabolism of hemoglobin (Hb), it is expected that bilirubin concentration will increase as Hb concentration increases. The Hb concentration and red blood mass also explain the diferences in TSB serum levels observed between males and females. It would be important to analyse hematological parameters of the neonates since they also apear to change across the seasons.

Answer:

Thank you for your comment; we agree with you. We have reviewed our data, included hematocrit levels as potential predictors of TSB levels, and analyzed these data. We found that the hematocrit levels were not associated with significant seasonal variations or sex differences. We have added these results in Tables 1 and 2 and also included relevant statements in the Methods, Results, and Discussion sections of revised manuscript.

Page no. 3 (lines 77, 87, 88, 92, and 96–98); page no. 4 (line 133); page no. 6 (Table 1); page no. 8 (Table 2); page no. 13 (lines 281–283 and 285).

  1. The study should also have taken into account the analysis of direct and indirect bilirubin levels. It would bring new clues in understanding this seasonal changes.

Answer:

Thank you for your suggestion. We agree with your comment. However, in the present study, we only analyzed TSB levels because we generally do not measure direct and indirect bilirubin levels during routine bilirubin evaluation as part of the 4th-day check-up of normal neonates. We have added statements regarding this issue as a limitation of our study in the Discussion section.

Page no. 14 (lines 299–303).

  1. One of the most important factors that affect total serum bilirrubin (TSB) levels in the caucasian population is the presence of the (TA)7 alelle (Gilbert Syndrome). The frequency of the (TA)7 allele is 30–40% in Caucasians, while it is less frequent in the Japanese population, with the frequency being approximately 16%. It has been suggested that the GLY71Arg variant and other polymorphisms may contribute to Gilbert's syndrome in Asian patients. To date, published data confirm the presence of GS contributes to neonatal hyperbilirubinemia, resulting in an important confouder factor to be taken into account.

Answer:

Thank you for your suggestion. Indeed, genetic factors play important roles in the development of neonatal jaundice. However, in the present study, genetic markers, such as the (TA)7 allele or Gly71Arg variant, were not investigated because genetic testing was not routinely conducted in neonates with physiological or pathological jaundice. However, in accordance with your suggestion, we have added information regarding Gilbert’s syndrome, (TA)7 allele, and Gly71Arg in the Discussion section.

Page no. 13–14 (lines 285–296).

  1. It is always important to consider additional factors that may affect serum levels of bilirubin, which may generate new criteria for the diagnosis and treatment of physiological and non-physiological neonatal jaundice.

Answer:

Thank you for your comment. We agree with you. We have discussed this aspect in the Discussion section of the revised manuscript.

Page no. 13 (lines 264–265).